# SCALING WEISFEILER–LEMAN EXPRESSIVENESS ANALYSIS TO MASSIVE GRAPHS WITH GPUS

## ABSTRACT

The Weisfeiler–Leman (WL) test is a cornerstone for analyzing the expressiveness of Graph Neural Networks, but computing its stable coloring at scale has remained a bottleneck. Classical refinement algorithms are inherently sequential and, despite optimal asymptotic complexity, do not exploit modern massively parallel hardware. Moreover, the problem is P-complete, suggesting limited parallelizability in the worst case. We show that these theoretical barriers do not preclude practical scalability. We obtain a linear-algebraic view of stable colorings by reformulating WL refinement as repeated matrix–vector multiplications. Building on this, we introduce two key contributions: (i) a randomized refinement algorithm with tight probabilistic guarantees, and (ii) a batching method that enables the analysis of stable colorings on subgraphs while preserving global correctness. This approach maps directly to GPU-efficient primitives. In numerical experiments, our CUDA implementation delivers up to ∼24x speedups over classical CPU-based partition refinement and, for the first time, successfully computes stable colorings on web-scale graphs with over 30 billion edges, where CPU baselines time out or fail.

## 1 INTRODUCTION

Graph Neural Networks (GNNs) have emerged as powerful tools for processing graph-structured data, achieving state-of-the-art performance across diverse domains such as chemistry, biology, social networks, and recommender systems Kunegis (2013); Kauffman (1969); Davis & Hu (2011). A crucial aspect of GNNs is their expressiveness—their ability to distinguish structurally distinct graphs. To analyze GNN expressiveness, the Weisfeiler-Leman (WL) graph isomorphism test, particularly its first-order variant (1-WL), has become fundamental (Grohe, 2021; Morris et al., 2023).

The 1-WL algorithm iteratively partitions the nodes of a graph based on local neighborhood structures. At each iteration, nodes receive a color based on the multiset of colors of their immediate neighbors, effectively encoding local topological information. When no further refinement is possible, the *stable coloring* groups nodes sharing identical structural roles within the graph. Computationally, straightforward implementations of stable coloring have $O(mn)$ runtime complexity, where $n$ is the number of nodes and $m$ is the number of edges of the graph. The most efficient approaches run in $O(m \log n)$ time and are based on seminal partition refinement algorithms (Paige & Tarjan, 1987). While distributed variants have been explored for multicore CPUs (Blom et al., 2008; van Dijk & van de Pol, 2016), these methods remain inherently hard to map to massively parallel hardware. To date, no approaches tailored to GPU architectures exist for 1-WL, creating a critical bottleneck for scaling to massive graphs.

In this paper, we close this gap by exploiting a connection between 1-WL and model reduction in dynamical systems. The stable coloring of 1-WL is equivalent to an equitable partition (Godsil, 1997; Grohe, 2021), where nodes in the same block share identical neighborhood counts. Equitable partitions unify perspectives across fields: they coincide with probabilistic bisimulation in Markov chains (Larsen & Skou, 1991) and backward equivalence (BE) in linear dynamical systems (Squillace et al., 2024; Cardelli et al., 2016). Across these perspectives, the partition guarantees an invariance property: states in the same block evolve identically over time if initialized equally. By framing an equitable partition through the linear system $w(t + 1) = Aw(t)$, with $A$ the adjacency

matrix, turns stable coloring into a sequence of matrix–vector multiplications—operations that map seamlessly to optimized GPU kernels.

To illustrate the idea, let $w$ be an initial coloring vector assigning the same component to all nodes sharing a color. Multiplying by the adjacency matrix $A$ refines the coloring: nodes with different neighborhood structures receive different values. For instance, in a three-node graph where node 1 connects to nodes 2 and 3, starting from $w = (c, c, c)^T$ yields $Aw = (2c, c, c)^T$, which already separates node 1 from nodes 2 and 3. Using any consistent initialization, e.g., $w = (c', c'', c'')^T$, produces the same refinement, confirming that nodes 2 and 3 are 1-WL (and BE) equivalent. Iterating this process yields the stable partition, and crucially, it requires nothing beyond repeated matrix–vector multiplications.

Building on this property, we make two main contributions. (i) We develop a Monte Carlo algorithm that replaces the exact symbolic vector computations (only used for illustration in the example above) with efficient, parallelizable numerical approximations, equipped with tight probabilistic error guarantees. (ii) We design a batching scheme that enables the analysis on subgraphs while ensuring globally correct stable colorings. BE lets us compress each batch to a surrogate subgraph; we then merge the batch-wise partitions into a global partition and rebuild a single quotient graph for the next refinement step.

Stable coloring is P-complete and thus unlikely to be efficiently parallelizable in the worst case (Chen et al., 2012). Our algorithm also has higher worst-case complexity than classical single-threaded refinement. Nevertheless, by exploiting GPU-friendly linear algebra primitives and keeping GPU memory usage under control by leveraging batching, we achieve substantial practical scalability. In extensive experiments with a CUDA-based implementation, our approach obtains $\sim$24x speedups over classical CPU-based partition refinement. On massive graphs with over 30 billion edges, it is the only method to successfully complete execution, whereas competitors either timed out or failed.

## 2 RELATED WORK

**Weisfeiler–Leman and graph expressiveness.** The Weisfeiler–Leman (WL) algorithm, particularly the first-order variant (1-WL), is widely recognized as essential for analyzing the expressiveness of Graph Neural Networks (GNNs) Xu et al. (2019); Morris et al. (2019); Grohe (2021). The characterization of 1-WL and stable coloring in terms of invariance properties of dynamical systems was established in Cardelli et al. (2016), instead. The main focus of Cardelli et al. (2016) was not so much on characterization but on model reduction though. That is, the reduction of the original dynamical system to a smaller one which is equally expressive. It is this feature that we use here in our batching scheme: by replacing subgraphs of the original graph with smaller subgraphs, the batching scheme is capable of obtaining a stable coloring of the original graph.

**Classic partition refinement algorithms.** Equitable partitions can be computed via partition refinement (Grohe, 2021), which finds the coarsest refinement of an input partition such that nodes in the same block have the same number of neighbors in every block. Efficient partition refinement algorithms achieve $O(m \log n)$ complexity (Paige & Tarjan, 1987; Valmari & Franceschinis, 2010), and this has been proven to be asymptotically tight (Berkholz et al., 2013). However, these splitter-driven algorithms rely on irregular memory access patterns, which map poorly to GPU execution models due to weak memory coalescing and cache contention (Shi et al., 2018; van Eerd et al., 2023). This motivates the search for alternative formulations aligned with GPU kernels.

**Parallel partition refinement algorithms.** Computing probabilistic bisimulation is **P**-complete Chen et al. (2012), which implies that efficient polylogarithmic-time parallel algorithms are unlikely in the worst case (i.e., not in **NC** unless **P** = **NC**). Since the adjacency matrix of a graph can be interpreted as a transition matrix of a Markov chain (van Dijk & van de Pol, 2016), this hardness limitation extends to BE and 1-WL. Still, it does not preclude practical parallelization. Distributed and multicore CPU algorithms have been developed (Blom et al., 2008; van Dijk & van de Pol, 2016), but these approaches do not map well to GPU architectures (we will compare with these in Section 5). GPU-tailored algorithms have also been proposed for strong bisimulation (Wijs, 2015; Martens & Wijs, 2024), but strong bisimulation is itself **P**-complete (Balcázar et al., 1992), and ex-

pressing probabilistic bisimulation through it incurs an exponential blowup (Chen et al., 2012). This severely limits the applicability of strong-bisimulation algorithms to 1-WL stable coloring.

**Randomized partition refinement algorithms.** A Monte Carlo variant of partition refinement based on the Schwarz–Zippel lemma (Schwartz, 1980) has been explored in a general setting that subsumes linear dynamical systems (Argyris et al., 2023). Our approach differs in three key respects: (i) it leverages a linear-algebraic characterization of BE that reduces refinement to repeated matrix–vector multiplications, (ii) it is designed for GPU execution with probabilistic error bounds integrated into an iterative coarsening pipeline, and (iii) it combines with a batching strategy that preserves global correctness while bounding per-batch VRAM usage.

**GPU graph processing frameworks.** General-purpose GPU graph libraries such as Gunrock (Wang et al., 2017), GraphBLAST (Yang et al., 2022), and NVIDIA's cuGraph provide highly optimized kernels for traversal and classical vertex coloring. However, they do not implement WL stable coloring, and they typically assume that the full graph fits in GPU memory ("scale-up") without support for external memory management or batching ("scale-out"). Our work addresses this gap with a GPU-based stable coloring algorithm capable of handling graphs much larger than GPU memory via a correct-by-construction batching approach.

## 3 BACKGROUND

We consider a graph $G = (V, A)$ where $V = \{1, ..., n\}$ is the set of nodes and $A \in \mathbb{R}^{n \times n}$ is the adjacency matrix with entries denoted by $a_{i,j}$. A partition $H = \{B_1, B_2, ..., B_{n'}\}$ of $V$ is a collection of $n'$ pairwise disjoint sets (called blocks) such that $\bigcup_{1 \leq i \leq n'} B_i = V$.

BE defines a partition of nodes such that each of the nodes in a block has equal sum of edge weights going to nodes of each block. Formally, BE requires that

$$\sum_{k \in B'} a_{i,k} = \sum_{k \in B'} a_{j,k}, \tag{1}$$

for all blocks $B, B' \in H$ and for all nodes $i, j \in B$.

**Reduced graph construction.** Given a BE partition $H$, it is possible to construct the super-graph $G' = (H, A')$ resulting from the coarsening of $G$ where each macronode is a partition block. The aggregated adjacency matrix $A' \in \mathbb{R}^{n' \times n'}$ is defined as

$$a'_{B,B'} = \sum_{j \in B'} a_{i,j}$$

for all $B, B' \in H$ and an arbitrary but fixed $i \in B$. When matrix $A$ is the adjacency matrix of a graph, BE identifies nodes according to regular equivalence Lorrain & White (1971). Likewise, BE coincides in this case with 1-WL stable coloring (Grohe, 2021; Morris et al., 2023) because in this case (1) simplifies to: nodes $i$ and $j$ share the same number of neighbors in block $B'$.

We next introduce the notion of coarsest refinement of an initial partition. Formally we say that a partition $H$ is coarser than a partition $H'$ if for any $B' \in H'$ there exists a $B \in H$ such that $B' \subseteq B$.

**Definition 1** (Coarsest BE partition). A BE partition $H$ of a node set $V$ is said to be the coarsest BE partition iff $|H| \leq |H'|$ for every BE partition $H'$ of $V$.

As discussed in Section 1, The coarsest BE refinement of an initial partition can be computed in $O(m \log n)$ time by partition refinement (Paige & Tarjan, 1987), as is done, for example, for Markov chain lumping by probabilistic bisimulation (Derisavi et al., 2003; Valmari & Franceschinis, 2010).

The BE coarsest partition refinement algorithms described in the literature (Derisavi et al., 2003; Valmari & Franceschinis, 2010) operate by refining an initial nodes partition iteratively until the BE criterion is met. Every iteration takes a block defined as a "splitter", and the total weight of every node of the graph towards nodes contained in the splitter block is computed. Each block is then refined by splitting it into multiple blocks until all of the blocks in the partition only contain nodes that have the same edge weight towards the considered splitter; new blocks originated from this operation are added to a queue of splitters to be considered in the next iterations.

---

**Algorithm 1** Randomized BE (RBE)

---

**Require:** Adjacency matrix $A$, node set $V = \{1, ..., n\}$,
**Require:** an initial node partition $H$
**Require:** set of rational numbers $Q = \{q_1, \ldots, q_r\}$
  1: **set** $H' \leftarrow \emptyset$
  2: **while** $H \neq H'$ **do**
  3:     **pick** randomly $w \in Q^n$ that is symmetric on $H$
  4:     **compute** coarsest partition $H'$ on which
  5:             $z = Aw$ is symmetric and that refines $H$
  6: **end while**

---

While these algorithms are very efficient asymptotically, they do not lend themselves to be adapted the GPU execution model very well. Practically, the main bottleneck is the reliance on irregular memory accesses. Such accesses are present, for example, in cumulative weight computation of every node. A graph's adjacency matrix is generally irregular by the very nature of graph data because not every node has the same outgoing neighbors; hence, memory accesses may involve potentially locations which are far apart, see for instance Shi et al. (2018).

The GPU model favors regular memory accesses to contiguous memory addresses due to its in-flight threads. This enables the exploitation of the burst mode of the device's DRAM memory, allowing for coalescing of multiple accesses into a single transaction. As a result, this greatly diminishes the memory bandwidth required to serve multiple concurrent threads, preventing their starvation Kirk & Hwu (2012). Modern GPUs are equipped with caches to mitigate the cost of non-contiguous memory operations but it is still easy to saturate them when operating with data structures of considerable size, thus making the partition refinement algorithm inefficient on GPUs when confronted with large adjacency matrices van Eerd et al. (2023).

## 4 METHODS

In this section, we first introduce and prove the correctness of the randomized (Monte Carlo) algorithm for computing a BE using linear algebra primitives (matrix vector-multiplication) that GPUs handle efficiently. Then, we discuss the correctness of the algorithm computing BE with batched subgraphs, ensuring global correctness.

### 4.1 MONTE CARLO LINEAR ALGEBRA ALGORITHM FOR BE COMPUTATION

We start by recalling that BEs can be characterized as invariant sets of matrices.

**Theorem 1** (Cardelli et al. (2016)). *We call $x \in \mathbb{R}^n$ symmetric on $H$ if and only if $x_i = x_j$ for all $B \in H$ and $i, j \in B$. Then, $H$ is a BE if and only if for all $B \in H$ it holds that $Ax$ is symmetric on $H$ for any $x \in \mathbb{R}^n$ that is symmetric on $H$.*

Intuitively, any two BE-equivalent coordinates $i, j$ remain equivalent after applying $A$. More formally, if $x$ is such that $x_i = x_j$ for all $B \in H$ and $i, j \in B$, then the same holds for vector $Ax$, that is, $(Ax)_i = (Ax)_j$ for all $B \in H$ and $i, j \in B$. As anticipated earlier, this yields an interpretation in terms of invariant spaces. Indeed, $A(U_H) \subseteq U_H$, where $A(U_H)$ is the image of $A$ under $U_H$, with $U_H$ being the linear subspace of all vectors symmetric on $H$:

$$U_H = \{x \in \mathbb{R}^n \mid x_i = x_j \text{ if } i, j \in B \text{ for some } B \in H\}.$$

The above result leads to our Randomized BE (RBE) algorithm, shown in Algorithm 1, where the symmetry check is done on a vector $w$ chosen at random (line 5).

**Theorem 2.** *RBE computes the coarsest BE $H'$ refining $H$, with an error probability of at most $n/r$. Assuming that $A$ has $m$ non-zero entries, the time and space complexity of RBE can be bounded by $\mathcal{O}(nm)$ and $\mathcal{O}(m)$, respectively.*[1]

---

[1]To simplify presentation, we assume $n \leq m$, i.e., we consider non-degenerated graphs.

As anticipated earlier, the worst case complexity of RBE is inferior to the optimal splitter-based implementation Berkholz et al. (2013); Grohe (2021) whose time complexity is $\mathcal{O}(m \log(n))$.

We next sketch why Algorithm 1 is correct. Let us assume that $H$ is not a BE partition. Then, by Theorem 1, there exists a $w \in \mathbb{R}^n$ which is symmetric on $H$ such that $H'$, the coarsest partition which refines $H$ and on which $Aw$ is symmetric, has more blocks than $H$. Continuing the argument, if also $H'$ is not a BE, by Theorem 1, there exists once again a $w' \in \mathbb{R}^n$ which is symmetric on $H'$ such that $H''$, the coarsest partition which refines $H'$ and on which $Aw'$ is symmetric, has more blocks than $H'$. Since $|V| = n$ and each further partition has at least one block more than the foregoing one, the process can continue up to $n$ many times. The fact that picking randomly a $w^{(i)}$ which is symmetric on $H^{(i)}$ yields with high probability a partition $H^{(i+1)}$ with the desired properties, instead, can be shown by means of the Schwarz-Zippel lemma Schwartz (1980).

Note that $n/r$ can be chosen very small in practice, e.g., in our experiments $r$ was $2^{64} > 10^{19}$ because $Q$ was the range of an unsigned 64 bit integer. Instead, the largest $n$ was below $10^{10}$, thus yielding an error probability of less than $10^{-9}$.

## 4.2 Batched BE Computation

The splitting for computing BE from batched subgraphs must be designed with care. Indeed, given an arbitrary edge batch $E_k$ (that is, a subset of the graph edges), deciding whether two nodes $i, j$ are BE-equivalent according to (1) can only be done if all outgoing edges of $i$ and $j$ are contained in $E_k$ (and we will call $i$ and $j$ *inner nodes*). If it happens that in the original graph a node with edges in batch $E_k$ has also an edge in batch $E_{k'}$ towards a node $j'$, than we call $i$ a *boundary node*. Each batch aggregates only inner nodes; every boundary node is treated as a singleton block. This guarantees correctness of the reconstructed global partition (see Theorem 3 below).

The intuition above is formalized in the following.

**Definition 2.** For a graph $(V, A)$, the edge set $E \subseteq V \times V$ is given by the node pairs $(i, j)$ such that $(i, j) \in E$ if and only if $a_{i,j} \neq 0$. For an edge partition $\mathcal{E} = \{E_1, \ldots, E_{N_b}\}$ of $E$, we define the inner nodes of batch $E_k$ as

$$V_k = \{i \in V \mid \nexists (i, j) \in E \setminus E_k\},$$

where we have assumed without loss of generality that each node has at least one outgoing edge.[2] The boundary nodes $\mathcal{B}$ are those nodes which are not inner nodes, that is, $\mathcal{B} = V \setminus \bigcup_k V_k$.

Note that $V_1, \ldots, V_{N_b}$ are pairwise disjoint since any node in $V_k$ has only outgoing edges in $E_k$.

With this definition in place, we can describe Algorithm 2 in more detail. The algorithm first splits the edges of the original graph into multiple batches (of size $M_b \geq 1$). This generates subgraphs $(V, A_{|E_k})$, where $A_{|E_k}$ is the restriction of matrix $A$ to the edge batch $E_k$. The algorithm computes a BE $H_k$ for each subgraph $(V, A_{|E_k})$ in a way that allows one to reconstruct from $H_1, \ldots, H_{N_b}$ a BE partition $H$ of the original graph $(V, A)$. Specifically, with $V_k$ being as in Definition 2, the algorithm computes $H_k$, the coarsest BE of $(V, A_{|E_k})$ which refines the initial partition $H_k^0$ from line 10. Partition $H_k^0$ ensures that only nodes whose outgoing edges are in $E_k$ (stored as $V_k$) can be aggregated together; this is necessary because the BE condition (1) can be only evaluated for nodes in $V_k$ — recall that we only have $A_{|E_k}$ at our disposal. Then, the union of $H_k$ blocks added in line 12 gives precisely $V_k$. Combining this with the computation of $H_k$ via $H_k^0$ ensures that $H$ in line 15 is indeed a BE partition. At the end of the repeat iteration, the algorithm checks if a further iteration should be conducted. This is the case if the current partition led to a reduction ($|V| < n$) or there is more than one batch ($N_b > 1$); in the $N_b=1$ case, the algorithm has run on the whole graph in that round, making an additional outer iteration unnecessary.

**Theorem 3.** *Fix a directed weighted graph $(V, A)$ and an edge partition $\mathcal{E}$ of $E$. If $H$ is a partition computed by Algorithm 2 in line 15, then $H$ is a BE of $A$. The time and space complexity of RBBE can be bounded by $\mathcal{O}(n^2 m)$ and $\mathcal{O}(m)$, respectively.*

The above result implies in particular that Algorithm 2 returns a BE of the original graph that can either be not aggregated further using BE or whose number of edges does not exceed the chosen batch size $M_b$. Note that the final partition is independent of batch order: each batch induces a

---

[2]If not, consider $A + I$ instead of $A$. It can be shown that $A$ and $A + I$ enjoy the same set of BE partitions.

---

**Algorithm 2 Randomized Batched BE (RBBE)**

---

**Require:** adjacency matrix $A$, node set $V$
**Require:** maximum number of edges in a batch $M_b \in \mathbb{N}$
 1: **repeat**
 2:     $H \leftarrow \{\}$
 3:     $E \leftarrow \{(i, j) \mid a_{i,j} \neq 0\}$
 4:     $n \leftarrow |V|$
 5:     $m \leftarrow |E|$
 6:     $N_b \leftarrow \lceil m/M_b \rceil$
 7:     **partition** $E$ in batches $E_1, \ldots, E_{N_b}$ with $|E_k| \leq M_b$
 8:     **compute** $V_1, \ldots, V_{N_b}$ according to Definition 2
 9:     **for** $1 \leq k \leq N_b$ **do**
10:         $H_k^0 \leftarrow \{\{V_k\}\} \cup \{\{i\} \mid i \in V \setminus V_k\}$
11:         $H_k \leftarrow \mathbf{RBE}(V, A_{|E_k}, H_k^0)$
12:         $H \leftarrow H \cup \{B \in H_k \mid B \subseteq V_k\}$
13:     **end for**
14:     $\mathcal{B} \leftarrow V \setminus \bigcup_{B \in H} B$
15:     $H \leftarrow H \cup \{\{i\} \mid i \in \mathcal{B}\}$
16:     $A \leftarrow \mathbf{AggregateAdjMatrix}(A, H)$
17:     $V \leftarrow H$
18: **until** $N_b = 1 \vee n = |V|$

---

monotone local refinement, we merge by the meet (intersection) on partitions, and the resulting fixed point is unique with high probability.

## 5 NUMERICAL EXPERIMENTS

### 5.1 EVALUATION SETUP

We evaluate our approach along three axes: (i) *reduction power*, i.e., how close the obtained partitions are to the coarsest equitable partition; (ii) *runtime efficiency*, measured against state-of-the-art CPU and GPU baselines; and (iii) *scalability*, focusing on graphs that exceed VRAM capacity.

**Datasets.** We use twelve medium- to large-scale graphs from established collections, including Flickr, Yelp, and Reddit from the PyTorch Geometric dataset collection (Fey & Lenssen, 2019), and nine web graphs from (Boldi & Vigna, 2004; Boldi et al., 2004; 2011; 2014). These cover sizes from fewer than $10^5$ nodes up to hundreds of millions of nodes and billions of edges. For scalability experiments, we additionally consider three massive graphs (gsh-2015, clueweb12, uk-2014) with more than 30 billion edges.

**Implementation.** Our algorithms are implemented in C++ with CUDA for GPU kernels and OpenMP for multicore parallelism. Matrix–vector multiplications are realized using device-wide reduction primitives from the CUB library, while partition canonicalization is handled on the CPU for efficiency. Random vectors are generated with 64-bit pseudorandom number generators (xoshiro256**, SplitMix64), ensuring negligible error probability ($< 10^{-9}$). Additional engineering details, including memory management and kernel optimizations, are deferred to Appendix B.

**Baselines.** We compare with (i) the partition refinement algorithm of Valmari & Franceschinis (2010) implemented as a single-threaded CPU program (denoted by **PR** in the results), (ii) our own RBE algorithm executed on a single CPU without batching (**RBE**), (iii) multicore probabilistic bisimulation reduction methods (Blom et al., 2008; van Dijk & van de Pol, 2016) using 32 and 64 cores (**SR**), and (iv) a Python implementation of the stable coloring algorithm using the WLConv layer in Pytorch (`torch_geometric.nn.conv.WLConv`), executed on CUDA device (**PT**).

**Hardware.** All experiments were conducted on a server with 8 NVIDIA H200 GPUs (141 GB VRAM each) and 2 TiB of RAM, running Ubuntu 24.04. Detailed hardware and software configurations are provided in Appendix C.

| | | | Randomized Batched BE (RBBE) | | | |
|---|---|---|---|---|---|---|
| *No.* | *Dataset Name* | *Size* | *100%* (coarsest) | *75%* | *50%* | *25%* |
| 1 | flickr | $n$: 89,250 $m$: 899,756 | 90.91% | 91.25% | 91.26% | 91.80% |
| 2 | cnr-2000 | $n$: 325,557 $m$: 3,216,152 | 26.24% | 26.24% | 26.24% | 26.24% |
| 3 | yelp | $n$: 716,847 $m$: 13,954,819 | 92.47% | 94.01% | 94.73% | 96.24% |
| 4 | ljournal-2008 | $n$: 5,363,260 $m$: 79,023,142 | 80.09% | 80.20% | 80.38% | 80.56% |
| 5 | hollywood-2009 | $n$: 1,139,905 $m$: 113,891,327 | 53.28% | 53.28% | 53.28% | 53.28% |
| 6 | reddit | $n$: 232,965 $m$: 114,615,892 | 93.45% | 94.49% | 94.85% | 96.07% |
| 7 | enwiki-2024 | $n$: 6,790,971 $m$: 172,762,484 | 91.20% | 91.41% | 91.52% | 91.78% |
| 8 | eu-2015-host | $n$: 11,264,052 $m$: 386,915,963 | 39.50% | 39.50% | 39.50% | 39.68% |
| 9 | arabic-2005 | $n$: 22,744,080 $m$: 639,999,458 | 29.88% | 29.88% | 29.88% | 30.07% |
| 10 | it-2004 | $n$: 41,291,594 $m$: 1,150,725,436 | 30.00% | 30.00% | 30.00% | 30.44% |
| 11 | twitter-2010 | $n$: 41,652,230 $m$: 1,468,365,182 | 86.92% | 87.77% | 87.79% | 87.85% |
| 12 | sk-2005 | $n$: 50,636,154 $m$: 1,949,412,601 | 29.95% | 29.95% | 29.95% | 30.20% |

Table 1: Batching (expressed in $\%M_b$) preserves reduction power: gap to the stable coloring (corresponding to the batch-free case $\%M_b = 100\%$) is below 5%.

**Metrics.** We report (i) the size of the final partition (i.e., the number of stable colors) relative to the number of nodes of the input graph (reduction power), (ii) wall-clock runtime excluding I/O, and (iii) success or failure within a fixed timeout (3600 seconds for medium-scale, 10,800 seconds for large-scale). For our RBBE implementation, we also evaluate different batch sizes, reported in terms of $\%M_b$, i.e., the relative to the total number of edges of the original graph $m$ ($\%M_b = 100\%$, 75%, 50%, 25%), to study the trade-off between memory usage and runtime.

**Parallel execution and device usage.** Unless stated otherwise, GPU experiments of our RBBE implementation use multiple GPUs. Edge batches are scheduled concurrently across devices and merged on the host between iterations. Crucially, when $\%M_b=100\%$ we have a single batch ($N_b=1$), so execution is effectively single-GPU; for smaller $\%M_b$ we obtain multiple batches, enabling multi-GPU concurrency by using a different device for each batch. Thus, $\%M_b$ influences not only per-batch cost but also device occupancy.

## 5.2 RESULTS

**Reduction power.** Table 1 reports the reduction power of RBBE at different batch sizes. Across all datasets, the obtained partitions are within 5% (in number of blocks) of the size of the partition obtained with $\%M_b = 100\%$. In this setting $N_b = 1$, so RBBE reduces to the batch-free RBE and returns the coarsest BE partition (i.e., the 1-WL stable coloring), with high probability. The results

| | CPU | | | | | GPU | | | |
| | Single core | | Multicore | | | Randomized Batched BE | | | |
| No. | PR | RBE | $SR_{32}$ | $SR_{64}$ | PT | 100% | 75% | 50% | 25% |
|---|---|---|---|---|---|---|---|---|---|
| 1 | 0.15 | 0.02 | 0.60 | 0.66 | 36.22 | 0.02 | 0.10 | 0.08 | 0.11 |
| 2 | 0.27 | 1.37 | 8.92 | 15.09 | T/O | 0.08 | 0.10 | 0.10 | 0.15 |
| 3 | 4.60 | 0.51 | 19.56 | 13.30 | 390.76 | 0.29 | 0.71 | 0.91 | 0.98 |
| 4 | 34.56 | 12.20 | 181.71 | 175.15 | T/O | 2.23 | 8.65 | 10.11 | 11.17 |
| 5 | 21.57 | 1.40 | 109.61 | 67.63 | 476.20 | 1.05 | 2.17 | 2.16 | 2.11 |
| 6 | 27.16 | 1.66 | 86.86 | 55.28 | 97.90 | 0.99 | 4.29 | 3.37 | 4.64 |
| 7 | 91.74 | 19.89 | N/A | N/A | N/A | 3.80 | 14.48 | 10.67 | 14.7 |
| 8 | 62.80 | 53.97 | N/A | N/A | N/A | 4.04 | 6.77 | 6.93 | 18.93 |
| 9 | 72.01 | 669.56 | N/A | N/A | N/A | 12.55 | 16.76 | 16.43 | 38.99 |
| 10 | 143.79 | T/O | N/A | N/A | N/A | 78.09 | 69.60 | 82.03 | 114.81 |
| 11 | 935.14 | 265.16 | N/A | N/A | N/A | 46.83 | 257.57 | 178.88 | 181.75 |
| 12 | 250.05 | T/O | N/A | N/A | N/A | 60.52 | 71.58 | 57.75 | 137.51 |

Table 2: Runtime comparison (wall clock execution in s); best highlighted in green. PR: single-thread partition refinement; RBE: proposed approach on a single CPU; $SR_{32}$/$SR_{64}$: multicore bisimulation, 32/64 cores; PT: Pytorch with WLConv layer; RBBE: proposed approach on GPUs for different batch sizes. RBBE 100% uses 1 GPU; RBBE 75/50/25% use 8 GPUs. RBBE 100% (1 GPU) achieves up to ∼24x over PR/RBE-CPU; RBBE 75/50/25% remain competitive while enabling concurrency. On overlapping datasets, $SR_{64}$ is 33–190x slower than RBBE on 1 GPU.

demonstrate that batching preserves partition quality while enabling GPU scalability. This confirms that the batching strategy does not significantly degrade the effectiveness of the reduction.

**Efficiency.** Table 2 reports runtime comparisons against the baselines. Interestingly, already on CPUs, RBE outperforms the classical refinement baseline (PR) on several datasets despite its higher asymptotic complexity, showing that the randomized linear-algebraic formulation can be advantageous even before GPU acceleration is applied.

The GPU-based RBBE implementation consistently outperforms both classical partition refinement (PR) and our own randomized algorithm executed on CPU (RBE), achieving up to ∼24x speedups, showing the advantages of hardware parallelization.

On all datasets where multicore probabilistic bisimulation reductions SR ran, our RBBE (100% batch) is 33x–190x faster than $SR_{64}$. These results suggest that multicore partition-refinement approaches alone do not bridge the performance gap for WL-style computations, whereas our linear-algebraic formulation maps efficiently to GPUs.

The straightforward PyTorch implementation of 1-WL (PT), obtained by repeatedly applying the WLConv layer until convergence, is easy to code but performs poorly in practice. It is consistently the slowest across all datasets (often by two orders of magnitude). Despite the easy accessibility of GPU execution in ML frameworks, their generic kernels are not necessarily suited to the irregular, iterative nature of stable coloring.

**Ablation on batch size under multi-GPU scheduling.** A smaller $\%M_b$ typically results in a batch that is cheaper to compute since fewer edges are considered but in turn it increases the overhead of an iteration as it results in a higher number of batches $N_b$ to consider. When using a multi-GPU setup it is possible to mitigate this overhead by distributing the batches of an iteration over multiple devices. This explains instances like sk-2005 (no. 12) where the $\%M_b = 50\%$ outperforms both the 75% and the 100% cases: more batches of smaller size are cheaper to compute and they can be handled concurrently on multiple GPUs; by contrast, the $\%M_b = 100\%$ column can exploit only a single GPU ($N_b$=1). We observe that the two datasets where smaller batch sizes yield the best runtime are precisely those with the strongest reduction (smallest ratio, no. 10 and 12).

| | | $M_b = \mathbf{1.8B}$ | | $M_b = \mathbf{4.5B}$ | |
|---|---|---|---|---|---|
| *Dataset* | *Size* | *Red. Pow.* | *Time* | *Red. Pow.* | *Time* |
| gsh-2015 | $n$: 988,490,691 $m$: 33,877,399,152 | 39.02% | 2956 | 35.84% | 1539 |
| clueweb12 | $n$: 978,408,098 $m$: 42,574,107,469 | 46.45% | 3788 | 43.03% | 2625 |
| uk-2014 | $n$: 787,801,471 $m$: 47,614,527,250 | 42.82% | 4592 | 37.87% | 3769 |

Table 3: Reduction power and wall-clock execution time (in s) on massive graphs (8 GPUs). Only our RBBE algorithm completes within resource limits; all CPU baselines (PR, RBE-CPU, SR) either timed out or ran out of memory. Two batch sizes have been considered: 1.8 and 4.5 billion edges.

We attribute this to a compounding effect of multi-GPU concurrency and early contraction: smaller batches increase the number of concurrent tasks ($N_b$), so the first round collapses large portions of the graph in parallel; the resulting aggregated graph is substantially smaller in subsequent iterations, thus amortizing merge overhead. Indeed, the right inset validates this assumption on dataset sk-2005 (no. 12) where we additionally compare with a run of RBBE with $\%M_b = 50\%$ case on single GPU (middle histogram). As expected, the wall-clock execution time is slower than the batch-free case (left histogram) due to the batching overhead (all batches must be processed sequentially); so the

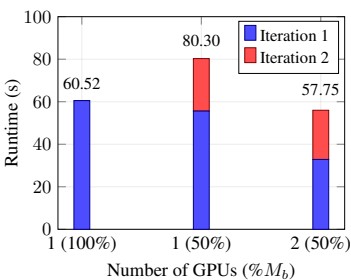

iteration time is essentially the *sum* over batch times. Instead, when two batches within the first iteration can be processed in parallel (right histogram), the wall-clock time of that iteration is the *maximum* time across such batches. This leads to an overall execution of the algorithm comparable to the batch-free case.

**Scalability to massive graphs.** Table 3 reports two per-device batch caps ($M_b = 1.8B$ and $M_b = 4.5B$ edges), both on 8 GPUs. In all three web-scale graphs (33–48B edges), our method is the only one to complete; CPU baselines time out or OOM. Increasing the cap from 1.8B to 4.5B (the maximum that fits in VRAM) improves both quality and runtime: final partitions are smaller (larger reduction power) and runtimes drop by about 1.2–1.9×. At this scale, the number of batches per round remains large in both settings; the larger cap mainly reduces cross-batch boundary effects and merge/synchronization overhead, and accelerates early contractions. These results suggest a rule of thumb for web-scale runs: use the largest per-device batch that fits in VRAM; when memory is tighter, smaller caps still complete—just with slightly weaker reduction and longer wall-clock.

To our knowledge, this is the first demonstration of stable coloring on graphs of this size, showing that GPU-tailored linear-algebraic formulations can overcome the bottlenecks of both sequential refinement and multicore CPU parallelism.

## 6 CONCLUSION

We introduced a GPU-amenable reformulation of the 1-WL stable coloring problem, grounded in a linear-algebraic view of equitable partitions and backward equivalence. By combining randomized matrix–vector refinement with a batching scheme that preserves global correctness, our approach achieves both theoretical guarantees and practical scalability. In extensive experiments, it outperforms classical refinement and multicore baselines by large margins and, for the first time, computes stable colorings on web-scale graphs with tens of billions of edges. More generally, the connection to dynamical-system reductions suggests new opportunities for importing linear-algebraic GPU methods into problems traditionally treated with inherently sequential algorithms.

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

## A  Appendix

### Proofs

*Proof of Theorem 2.* Correctness was addressed in the main text, using the existence of a vector $w$ that a) is symmetric on the current partition and b) can be used to refine the current partition. The actual algorithm, instead, picks randomly a $w$ which is symmetric on the current partition, hoping that this will yield a vector that splits the current partition with high probability. We next outline why this is indeed the case. To this end, let us assume that we are given a partition $H$ which is not a BE. It is possible to construct a matrix $M \in \mathbb{R}^{\kappa \times n}$ such that $My = 0$ if and only if $y$ is symmetric on $H$, where $\kappa = n - |H|$ (notice, for instance, that $w_1 = w_2$ is equivalent to the linear equation $w_1 - w_2 = 0$). Moreover, let $L \in \mathbb{R}^{n \times H}$ be the matrix such that $(Lz)_i = z_B$ for any $z \in \mathbb{R}^H$, $B \in H$ and $i \in B$. Notice in particular that there is a one-to-one relation between $\mathbb{R}^H$ and vectors in $\mathbb{R}^n$ that are symmetric on $H$. Since $H$ is not a BE, it holds that $MAL \neq 0$. Indeed, there must exist a $w$, symmetric on $H$, such that $MAw \neq 0$. For any such $w$, one can find the corresponding $z$ with $Lz = w$, yielding $MALz \neq 0$. What are the chances that $MALz = 0$ for a randomly picked $z \in \mathbb{R}^H$? Obviously, the kernel of $MAL$ is a proper subset of $\mathbb{R}^H$ because $MAL \neq 0$. Hence, the kernel is a Lebesgue zero set in $\mathbb{R}^H$, meaning that a randomly chosen point in $\mathbb{R}^H$ will be almost surely not in the kernel. To allow for an algorithmic treatment, we pick numbers from $Q$. Each row of $MAL$ defines a polynomial of degree one in variables $z_{B_1}, \ldots, z_{B_{|H|}}$. Hence, $MALz = 0$ if and only if $z$ is a root of $n - |H|$ degree one polynomials. Allowing $z$ to attain values in $Q$, the Schwarz-Zippel lemma Schwartz (1980) ensures that the probability of randomly picking a root $z$ of a degree one polynomial is at most $1/|Q| = 1/r$. Hitting a root of $n - |H|$ polynomials at the same time comes with an even smaller probability. However, since $MAL$ might have linear dependencies, we use the bound $1/r$ for simplicity. Since the original partition can be refined at most $n$ times, the error probability can be bounded by $n/r$. In the last part, we consider the worst case complexity of the algorithm. Again, since there can be at most $n$ refinements, and each partition refinement is obtained by means of a matrix-vector product, the number of steps can be bounded by $\mathcal{O}(nm)$, while the space constraints amount to $\mathcal{O}(m)$, where $m$ is the number of non-zero entries of $A$. Since partitions can be compactly stored as vectors of length $n$ and the coarsest partition on which two given vectors are symmetric can be computed in linear time, this completes the proof.

$\square$

*Proof of Theorem 3.* Let us assume that $H$ has been computed by Algorithm 2 in line 15. To show that $H$ is indeed a BE, pick any $B, B' \in H$ and $i, j \in B$. We need to show that

$$\sum_{k \in B'} a_{i,k} = \sum_{k \in B'} a_{j,k} \tag{2}$$

In the case $B$ contains a boundary node, $B$ must be a singleton block (as boundary nodes are added in line 15) and there is nothing to prove because $i = j$. Let us thus assume that $B$ has no boundary nodes. In this case, block $B$ was add to $H$ in line 12. Hence, there is a unique $\kappa$ such that $i, j \in V_\kappa$ and

$$\sum_{k \in B': (i,k) \in E_\kappa} a_{i,k} = \sum_{k \in B': (j,k) \in E_\kappa} a_{j,k}$$

Here, we implicitly assumed that the result of the randomized Algorithm 2 was indeed correct, an event whose probability can be made arbitrarily large by Theorem 2. Moreover, due to the construction of $V_\kappa$, for any $S \subseteq V$, it holds that

$$\sum_{k \in S} a_{i,k} = \sum_{k \in S: (i,k) \in E_\kappa} a_{i,k}.$$

A combination of the last two statements yields (2).

For the proof of complexity, we first estimate the complexity of one until loop iteration. To this end, we note that batches $E_1, \ldots, E_{N_b}$ can be computed in $\mathcal{O}(m)$ time and space. Likewise, $V_1, \ldots, V_{N_b}$ can be computed in $\mathcal{O}(m)$ and $\mathcal{O}(n + m)$ space by adding for each edge $(i, j) \in E$ its batch label $k$ (when $(i, j) \in E_k$) to the node it originates from, i.e., we add $k$ to a list associated to node $i$; after iterating through all edges, the inner nodes are these which have exactly one batch label. For the complexity of the for loop, we first note that the for loop comprises $N_b$ invocations of RBE for matrices with at most $M_b$ non-zero entries. This gives, respectively, a time and space complexity of $\mathcal{O}(nm)$ and $\mathcal{O}(n + m)$ by Theorem 2 because $\sum_k |E_k| = m$. Additionally to that, the updates of $H_k^0$ and $H$ yield time complexity $\mathcal{O}(n)$, which is not dependent on $N_b$. This can be realized by storing partitions as vectors and by updating only the part of the vector that is required in the current for loop iteration. For instance, each $H_k^0$ can be constructed by replacing $|V_k|$ entries of the vector $(1, 2, 3, \ldots, n)^T$ which encodes partition $\{\{i\} \mid 1 \leq i \leq n\}$. A similar remark applies to updates of partition $H$. Since the aggregated matrix can be computed in $\mathcal{O}(nm)$ steps (for each block representative $i_B$, find all its outgoing edges in $A$ and sum them according to the blocks of partition $H$), the time and space complexity of one until loop iteration can be overall bounded by $\mathcal{O}(nm)$ and $\mathcal{O}(m)$, respectively. Since there are at most $n$ iterations of the until loop (this occurs if every batching reduces the graph by exactly one node), we obtain the complexity statement. $\square$

## B    IMPLEMENTATION

Both Algorithms 2 and 1 are implemented in C++ with the CUDA toolkit Nickolls et al. (2008). Algorithm 2 is a CPU algorithm while the inner iteration of Algorithm 1 has GPU sections; batch iterations in Algorithm 2 are executed in parallel using the OpenMP library to exploit all the GPUs available in the system Dagum & Menon (1998). At the start of Algorithm 1, initial partition $H_k^0$ is transferred as a vector $y$ to the GPU memory, where nodes $i, j$ are in the same block of $H_k^0$ if and only if $y_i = y_j$; instead, the matrix-vector product of Algorithm 1 is carried out using the device-wide reduction primitives of the CUB library. A custom CUDA kernel is used in line 3 where entries of $w$ are computed as random entries according to the entries of $y$. Specifically, random entries are obtained by applying the `xoshiro256**` random number generator to the output of a `SplitMix64` generator with $y$ as the vector of seeds, i.e.

$$w_i = \texttt{xoshiro256**}(\texttt{SplitMix64}(y_i))$$

for all $i$, where each $w_i$ is stored as an unsigned 64 bit integer (thus yielding $m = 2^{64}$ in Algorithm 1), see Blackman & Vigna (2021); Steele et al. (2014) for more information on the routines. Line 2 of Algorithm 1 is realized by storing partition $H'$ as vector $z = Aw$ and by comparing first the number of unique elements of $z$ and $w$. This is a necessary but not sufficient condition for the equivalence in line 2. This first preliminary check is carried out on GPU memory using the device-wide unique primitive of the CUB library. If the check is passed the two vectors are transferred to the

host memory where their canonical form is computed using a sequential algorithm optimized with the use of the Abseil flat hash map implementation Winters (2018). This two-step process allows the exploitation of the CPU for the computation of the canonical forms, which is a mostly sequential operation. At the same time, it minimizes the number of times the two arrays are transferred from the GPU.

## C    USE OF LLMS

LLMs have been used during the writing of this paper only for the purpose of polishing the text and the grammar. All the ideas, the algorithms and their implementations where made solely by the authors without the use of LLMs.

