# OpenReview forum: "Scaling Weisfeiler–Leman Expressiveness Analysis to Massive Graphs with GPUs"
_ICLR.cc/2026/Conference — Submitted to ICLR 2026_

### Official Review · Reviewer_vpTt · 2025-10-20

**Soundness:** 1
**Presentation:** 1
**Contribution:** 3
**Rating:** 2
**Confidence:** 4

**Summary:**

This paper proposes a probabilistic WL variant that can be parallelized and efficiently scales to very large graphs.

**Strengths:**

**(S1) Theoretical contribution (Originality, Significance):** Parallelizing the inherently sequential WL algorithm is a significant and interesting contribution.

**(S2) Empirical results (Quality):** Efficiently scaling WL to billions of nodes is impressive. In particular, the speed-up over classical WL seems significant. The experiments demonstrate that (1) the proposed batched algorithm is faster than previous algorithms; and (2) that it scales to very large graphs.

**Weaknesses:**

**(W0) Minor weakness:** This work could be clearer about the applications of parallelizing WL. While WL is used to evaluate GNN expressivity, it is mainly used as a mathematical tool for proofs rather than as a practical algorithm. While WL is used practically in graph kernel methods, I am unaware of any recent works using these methods.

**(W1) Clarity:** While the underlying concept of the WL algorithm is intuitively simple, the paper is difficult to follow. In particular:

 **(W1.1)** The connection to connection to backward equivalence (BE) in dynamical systems is unclear. Defining these concepts and showing an example would help (the example in line 57 does not explan the connection to BE).

> BE defines a partition of nodes such that each of the nodes in a block has equal sum of edge weights  going to nodes of each block. Formally, BE requires that (...)

This explains what BE defines, but does not **define** BE (see also W2). Are these all properties of a BE?

**(W1.2)** The P-completeness of stable coloring does not follow from the explanation. As this is a key motivation behind the paper it is important that this gets communicated clearly.

**(W2) Formality:**  Definitions and Theorems are incomplete or not formal enough. Consider Theorem 1:

> We call $x \in \mathbb{R}^n$ symmetric on $H$ if and only if $x_i = x_j$ for all $B \in H$ and $i, j \in B$. Then, $H$ is a BE if and only if for all $B \in H$ it holds that $Ax$ is symmetric on $H$ for any $x \in \mathbb{R}^n$ that is symmetric on $H$.

What is $H$? Only from previous context one can conclude that $H$ is a partition of a graph (which is not part of this definition). What is $A$? It is probably the adjacency matrix of the graph for which $H$ is the partition of. However, the reader should not need to guess what a definition or theorem means. Finally, in the second sentence, the authors quantify over all $B \in H$ but then never use $B$.

This (W2) is my key issue. The papers formalization are incomplete or difficult to follow, making it difficult to evaluate the correctness.

**(W3) Experimental results:** While the explanation of experiments is difficult to follow, I have multiple problems with the implication of the results.
- In Table 2, the fastest model  is in 10 / 12 cases the single-GPU model raising questions about the practical usefulness of batching WL. Even on the large graphs of Table 3 (with > 1 billion nodes), larger batch sizes are faster than smaller ones. Thus, batching only seems to be useful if the graphs are too large to fit into memory. How much VRAM each graph dataset requires.
- The naive PyTorch 1-WL implementation is significantly slower than I would intuitively expect. Do the authors have an explanation for this? I quickly checked the code but could not find the code for the WL baseline.
- The massive graph experiments (Table 3) lack CPU baselines that completed, making it impossible to quantify speedups at this scale.

**Questions:**

- What is AggregateAdjMatrix in Algorith 2?
- Please see weaknesses

#### Other Pointers
- Table 1 needs more than half a page and contains little information.
- The graph definition should state that it is weighted graph.
- Refinement is not formally introduced (furthermore, the concept of BE refinement seems different than that of WL refinement).
- This paper could be significantly improved by using figures to illustrate concepts such as definition or algorithms.
- Please make references to the appendix clickable (e.g. using \label and \ref)
- At least one figure without a caption.

---

### Official Review · Reviewer_ZLaf · 2025-11-01

**Soundness:** 3
**Presentation:** 3
**Contribution:** 3
**Rating:** 6
**Confidence:** 3

**Summary:**

The authors propose a reformulation of the WL test as a repeated-matrix vector operation. The approach can be efficiently mapped to a parallelized implementation on GPUs/ Tje stable coloring of 1-L is formulated as an iterative set of multiplications, some node coloring vector with the adjacency matrix.  The key contributions of the work are an MC algorithm that uses randomized vectors to find partitions with probabilistic guarantees and a batched algorithm to enable processing large graphs while preserving global correctness. The algorithm provides an order-of-magnitude speedup over even multicore CPU baselines. The algorithm substitutes time complexity for parallelizability.  The authors evaluate reduction power and runtime efficiency on 12 medium to large-scale graphs and scalability studies on 3 massive web-scale graphs and show strong results over baselines.

**Strengths:**

- The proposed methodology is sound, and a trade-off for worse time complexity for practical improvements is well justified
- While the reformulation only provides probabilistic bounds, the trade-off is negligible
- Evaluation on large graphs with tens of billions of edges is the largest evaluation done on this scale. Baselines are not able to

**Weaknesses:**

- The evaluation is a bit confusing at times. Table 1 may be alternatively presented as a percent change between the baseline and batched methods.
- RBBE enables concurrency but does not provide improvements in wall time after parallelization. The authors mention this is due to the merge overhead, but it would be good to provide performance numbers beyond a single dataset (sk2005) if possible. This can also inform strategies for parallelization.
-  The scope of the work is limited to 1-WL only and generalized k-WL is not discussed

**Questions:**

- The CPU baseline fails due to OOM, but wouldn't the batched implementation also run on CPU?

---

### Official Review · Reviewer_3iTk · 2025-11-06

**Soundness:** 2
**Presentation:** 2
**Contribution:** 1
**Rating:** 2
**Confidence:** 4

**Summary:**

This paper proposes a mult-gpu algorithm and cuda implementation for getting 1-WL coloring or the coarsest equitable partition. The paper claims designing two algorithms, the first is based on iterative refinement with a random starting vector via power iteration, which is suitable to find approximate coarsest equitable partition when converge for small graphs/subgraphs that can fit into single-gpu memory. The second algorithm is basically a way to split the giant graph into blocks of local graphs for iterative refinement, and then merge these local partitions back to the global partition of the giant graph. The author implemented them with Nvidia cuda libraries and tested its effectiveness in large-scale graphs.

**Strengths:**

1. The writing is clear, while I think the background knowledge is too much for the main paper, better to put into appendix.
2. Each designed algorithm has error analysis and theoretical support for finding coarsest equitable partition.
3. The speed improvement claimed by the paper is great.
4. The ability to handle billons of edges is very useful, given all existing baselines are not working.

**Weaknesses:**

1. The central idea of using power iteration to find equitable partition / 1-WL color refinement is not new at all. For example, Power-Iterated Color Refinement (AAAI’14): establishes the power-iteration lineage for refinement.  The connection between 1-WL and matrix vector production is well known in the commuity.
2. Also, using random evaluations to separate equivalence classes is a standard trick (Schwartz–Zippel-style arguments underpin many Monte-Carlo tests) — again, the probabilistic splitter idea isn’t brand new, even if the exact instantiation varies.
3. It seems to me that the main contribution is: 1) implementation on GPUs. 2) partition the massive graph into blocks while keeps boundary nodes as singletons for correctness. Nevertheless, I still feel that there is not much novelty inside. I think it would be interesting if the author can:
- study how to batching graphs comparing to the current random batch to make the merged partition much accurate. For now we can see from Table 3 that increasing the number of batches hurts the correctness a lot.

**Questions:**

1. Where the is the metric for the gap to oracle equitable partition? The current reduction power does not show that. Given that the algorithm relies on randomness with certain error analysis, it is important to show that empirically the real-world error is matching the theorem.

2. Can you talk more on how you merge the boundary nodes?

3. Is there any better batching split method and partition merging method other than random?

4. It seems that the code is working on 8GPUs at most? How about mult-node version?

---

### Official Review · Reviewer_Y6CY · 2025-11-09

**Soundness:** 3
**Presentation:** 3
**Contribution:** 2
**Rating:** 4
**Confidence:** 3

**Summary:**

The paper proposes a new computing scheme for 1-WL coloring by viewing refinement as repeated matrix–vector multiplications. Building on this, the authors introduce a randomised refinement algorithm with probabilistic guarantees and a batching scheme that operates on subgraphs while preserving global correctness. Experiments across social/web graphs show near-coarsest partitions under batching (≤5% gap) and up to 24× wall-clock speedups vs classical CPU baselines.

**Strengths:**

1 GPU-based RBBE implementation consistently outperforms classical CPU implementations.

2. The guarantee of error ≤ n/r makes the method practical for real-world scenarios.

3. The method can be scaled to multiple GPUs.

**Weaknesses:**

1. Today there are many frameworks that run GNNs efficiently such as Pytorch Geometric and Deep Graph Library. It is also known that the Graph Neural Network of GIN [1] has an expressive power as 1-WL. Therefore, I believe a comparison between an efficient GIN and the proposed method is required, where GIN will be used to compute the coarsest refinement.

2. Tables and results are very difficult to read.

3. The actual implementation is not based on real random number selection. How does it affect the empirical error and theoretical guarantees?


[1] Xu, Keyulu, et al. "How Powerful are Graph Neural Networks?." International Conference on Learning Representations, 2019

**Questions:**

Please see my questions above.

---

### Meta-Review · Area_Chair_HYgb · 2025-12-22

**Summary:**

This paper studies the computational scaling of the 1-dimensional Weisfeiler–Leman (1-WL) refinement, a tool for evaluating GNN expressiveness, to massive graphs via GPUs. The authors reformulate WL refinement as repeated matrix–vector multiplications, enabling a randomized parallel algorithm with probabilistic guarantees and a batched refinement scheme that handles graphs exceeding single-GPU memory while maintaining global correctness. A CUDA implementation shows large empirical speedups (up to ~24×) over classical CPU-based partition refinement and enables computation on web-scale graphs with tens of billions of edges where baselines fail.

Reviewers agreed that scaling WL to this regime is impressive and practically useful, particulary given CPU baselines timeout or OOM. However, opinions split on novelty, formal clarity, and experimental presentation. Some saw it as a strong systems contribution with solid theoretical grounding, others deemed conceptual ideas largely known and raised serious concerns about exposition, formal definitions, and evaluation clarity. The paper is not competitive.

**Reviewer Concerns:**

No rebuttal is provided by the authors.

Shared comments:

1. Novelty vs. Engineering Contribution. Multiple reviewers noted key conceptual ingredients—linear-algebraic views of 1-WL, power iteration, and randomized splitting of equivalence classes—are not entirely new, appearing in prior work.

2. Clarity and Formalization. Reviewers repeatedly flagged incomplete or unclear definitions (e.g., backward equivalence, refinement, theorem notation), missing formal context, and difficulty following presentation. This impeded evaluating correctness for some.

3. Evaluation and Metrics. Several found experimental tables hard to read, requesting clearer metrics like explicit gaps to oracle coarsest partitions, batching-induced error reporting, and batching trade-off justification. Questions also on why batching sometimes doesnt improve wall-clock time beyond single-GPU.

4. Scope and Practical Impact Some questioned practical relevance of scaling WL, noting its often theoretical use, and asked for clearer downstream applications.

Reviewer-Specific Questions:

1. Reviewer Y6CY: Requested comparisons with efficient GIN-based pipelines and randomness implementation clarification. Not addressed in rebuttal (no author response provided), highlighting paper positions as WL computation engine, not learning-based alternative.

2. Reviewer 3iTk: Skeptical of novelty, viewing main contribution as GPU implementation and batching. Raised detailed questions on batching correctness, merging boundary nodes, and empirical error vs. theoretical guarantees. Emphasizes need for clearer correctness diagnostics and batching rationale.

3. Reviewer ZLaf: Viewed work more positively, emphasizing probabilistic trade-off soundness and unprecedented evaluation scale. Raised minor concerns about presentation and scope (limited to 1-WL).

4. Reviewer vpTt: Acknowledged parallelizing WL significance and scaling results, but raised strong concerns about formal rigor, clarity, and experimental interpretation, ultimately recomendng rejection.

**Reviewer Scores:**

The scores of all reviewers are reasonable.

---

### Decision · Program_Chairs · 2026-01-26

Reject